# Downregulation of Exosomal hsa-miR-551b-3p in Obesity and Its Link to Type 2 Diabetes Mellitus

**DOI:** 10.3390/ncrna9060067

**Published:** 2023-11-02

**Authors:** Kseniia V. Dracheva, Irina A. Pobozheva, Kristina A. Anisimova, Stanislav G. Balandov, Maria N. Grunina, Zarina M. Hamid, Dmitriy I. Vasilevsky, Sofya N. Pchelina, Valentina V. Miroshnikova

**Affiliations:** 1Petersburg Nuclear Physics Institute Named by B.P. Konstantinov of National Research Centre “Kurchatov Institute”, 188300 Gatchina, Russia; 2Department of Molecular-Genetic and Nanobiological Technologies, Scientific Research Center, Pavlov First Saint-Petersburg State Medical University, 197022 Saint Petersburg, Russia; 3Center for Surgical Treatment of Obesity and Metabolic Disorders, Pavlov First Saint-Petersburg State Medical University, 197022 Saint Petersburg, Russia

**Keywords:** exosomal microRNAs, obesity, type 2 diabetes mellitus

## Abstract

Obesity is a significant risk factor for the development of type 2 diabetes mellitus (T2DM). Adipose tissue dysfunction can affect the pool of circulating exosomal miRNAs, driving concomitant disease in obesity. These exosomal miRNAs can reflect adipose tissue functionality, thus serving as prognostic biomarkers for disease monitoring in case of T2DM. In the present study, we conducted NanoString microRNA profiling of extracellular vesicles (EVs) secreted by adipose tissue of obese patients (body mass index (BMI) > 35) without T2DM and nonobese individuals (BMI < 30) as a control group. Functional and pathway enrichment analysis showed that miRNAs associated with obesity in this study were implicated in insulin signaling and insulin resistance biological pathways. Further, these microRNAs were screened in serum EVs in the following groups: (1) obese patients with T2DM, (2) obese patients without T2DM, and (3) nonobese individuals as a control group. has-miR-551b-3p was shown to be downregulated in adipose tissue EVs, as well as in serum EVs, of patients with obesity without T2DM. At the same time, the serum exosomal hsa-miR-551b-3p content was significantly higher in obese patients with T2DM when compared with obese patients without T2DM and may be a potential biomarker of T2DM development in obesity.

## 1. Introduction

Obesity is an escalating issue worldwide and, as a significant driving factor of type 2 diabetes mellitus (T2DM), contributes to the increase in the prevalence of T2DM [1]. Obesity-induced adipose tissue overaccumulation and remodeling is associated with the overproduction of metabolites and inflammatory cytokines, subsequent low-grade systemic inflammation, insulin resistance, β-cell dysfunction, the onset of hyperglycemia, and the eventual development of T2DM [2]. The current opinion is that most patients with obesity can go through a transitional stage called “prediabetes” before developing hyperglycemia, when the glucose levels are not high enough for a T2DM diagnosis [1]. Therefore, there is a need for relevant biomarkers for early diagnosis, progression monitoring, and targeted therapy of T2DM; circulating microRNAs could be the best choice [3,4]. However, there is still no consensus on the precise miRNA signature in different cohorts of patients with T2DM, and little is known about the functional roles of the identified miRNAs in metabolic processes within the link to T2DM pathogenesis.

miRNAs are non-coding RNAs of 19–22 nucleotides that function as negative regulators of translation and are involved in many cellular processes [5]. In blood, miRNAs are presumably packaged in microparticles like exosomes, microvesicles, and apoptotic bodies or associated with RNA-binding proteins (Argonaute 2) or high-density lipoprotein complexes, and this helps them to avoid degradation [6]. RNA sequencing analysis demonstrated that miRNAs prevail among the human plasma-derived exosomal RNA species [7].

Extracellular vesicles (EVs, i.e., exosomes, microvesicles) are present in all types of human body fluids, including serum [4]. These nano-sized membranous particles are produced and released by almost all cell types, and they play an essential role in cell-to-cell communications by delivering different bioactive compounds such as functional proteins, metabolites, and nucleic acids, including microRNA, to recipient cells [4]. EV biogenesis is believed to be disturbed in obesity, with consequent variations in their composition, including miRNAs [8,9]. In addition, the concentration of plasma EVs increases in obesity [10,11,12,13]. Recently, we provided the first evidence concerning the morphological changes in EVs derived from adipose tissue during obesity [14]. Moreover, studies demonstrated that the transfer of adipose-tissue-released EVs from ob/ob mice induced insulin resistance, adipose inflammation, and hepatic steatosis in lean mice [15,16]. Thus, profound effects on metabolic diseases, such as T2DM, of adipose tissue EVs in obesity have been suggested [3].

It is worth noting that adipose tissue is a main source of serum exosomal miRNAs, as mice with a fat-specific knockout of the miRNA-processing enzyme Dicer (ADicerKO) present major decreases in circulating exosomal miRNAs, as do humans with lipodystrophy [5]. At the same time, miRNAs are also present in the circulation outside of EVs, but the reduction in total miRNAs in ADicerKO serum is not so dramatic [5]. As we need effective biomarkers for monitoring T2DM development during obesity, when pronounced adipose tissue dysfunction takes place, serum exosomal miRNAs could be more informative.

To date, there have been several reports of altered circulatory miRNA signatures during diabetes [6,17,18]. Most studies on extracellular miRNA expression patterns in obesity involved the simple isolation of miRNAs from blood/plasma/serum [19], and only a few have studied EVs [20,21,22,23]. Our study aimed to find miRNAs that could reflect shifts in adipose tissue dysfunction in adipose tissue EVs in obesity. As primarily visceral adiposity is associated with metabolic changes, at the first stage, we studied the obesity-associated microRNA signature of EVs secreted by visceral adipose tissue (VAT) during its cultivation ex vivo. At the second stage, the abundance of microRNAs that demonstrated a different composition in VAT EVs in obesity was compared in serum EVs in obese patients depending on T2DM diagnosis.

## 2. Results

The demographic and anthropometric data characterizing the study cohort are presented in Table 1. The first stage of the research included studying the miRNA composition of adipose-tissue-derived EVs extracted from culture medium upon adipose tissue cultivation ex vivo. At the second stage, the content of obesity-associated microRNAs was estimated in serum EVs in obese patients depending on T2DM diagnosis.

We used commercial exosome isolation reagents for the rapid isolation of exosome-enriched EVs from rather small volumes of culture medium and serum for further determination of the miRNA content. To confirm their origin, the extracted EVs were analyzed via Western blot. Adipose tissue EVs contained the canonical exosomal markers CD63 and CD81, as well as adipose-tissue-specific proteins—fatty acid binding protein 4 (FABP4), omentin-1, and adiponectin. A representative Western blot is shown in Figure 1A. Serum EVs were tested for CD81 and for the specific marker FABP4 to show that the fraction included adipose tissue EVs (Figure 1B).

### 2.1. miRNA Profiling of Adipose Tissue EVs

At the first stage of the study, miRNA profiling of EVs secreted by obese VAT was performed using NanoString technology. The human v3 miRNA expression assay kit allows for the simultaneous detection of 800 miRNAs. Four obese patients without T2DM (1/3 male/female, BMI > 35) and four non-obese controls (1/3 male/female, BMI < 30) were included. The data analysis showed that three miRNAs—hsa-miR-1246, hsa-miR-145-5p, and hsa-miR-551b-3p—were downregulated in VAT EVs from obese individuals (*p* < 0.05) (Figure 2). Since SAT samples were only analyzed for obese patients, we were able to additionally compare the VAT and SAT miRNA profiles in obesity. Two miRNAs—hsa-miR-132-3p and hsa-miR-302d-3p—prevailed in SAT EVs, while hsa-miR-10a-5p was more abundant in VAT EVs (Appendix A).

Functional and pathway enrichment analysis was performed using the DAVID database. The analysis showed that the products of target genes of differently abundant miRNAs in adipose tissue EVs are involved in biological pathways associated with cellular lipid metabolism, differentiation of adipose tissue cells, insulin signaling, and insulin resistance pathways (GO:0044255, GO:0010887, GO:0045444; KEGG: hsa05417, hsa04910, hsa04931). Thus, one can assume that exosomal hsa-miR-1246, hsa-miR-145-5p, hsa-miR-551b-3p, hsa-miR-132-3p, hsa-miR-302d-3p, and hsa-miR-10a-5p could be linked to the development of metabolic complications in obesity, including T2DM. Thus, these microRNAs were further studied in serum EVs in enlarged cohorts of patients with obesity without T2DM and nonobese controls and additionally compared with a group of obese patients diagnosed with T2DM (Table 1).

### 2.2. miRNA Levels in Serum EVs in Obesity and T2DM

Among the obesity-linked adipose tissue exosomal miRNAs, a reduction in serum exosomal content in obese patients without T2DM compared with the control group was demonstrated only for hsa-miR-551b-3p (Figure 3). Interestingly, the level of hsa-miR-551b-3p in serum EVs in the T2DM group was higher than that in obese patients without T2DM (Figure 3). hsa-miR-302d-3p, which was shown to be more abundant in SAT EVs compared to VAT, also demonstrated an increase in serum EVs in the T2DM group, as compared to obese patients without T2DM (Appendix A). Additionally, it tended to be reduced in serum EVs of nondiabetic obese individuals, as compared to controls.

ROC analysis demonstrated moderate predictive efficiency of hsa-miR-551b-3p (Figure 4).

We determined whether the expression of exosomal miRNAs correlated with clinical parameters in the study cohorts. There was a positive correlation between the hsa-miR-551b-3p level in serum EVs and glycated hemoglobin (HbA1c) (r = 0.491, *p* = 0.028 in the entire cohort; r = 0.616, *p* = 0.005 in the combined obese cohort). In the T2DM subgroup, the hsa-miR-551b-3p level in serum EVs was negatively correlated with BMI (r = −0.732, *p* = 0.003). Patients without T2DM also showed a positive correlation between the serum exosomal hsa-miR-302d-3p content and triglyceride concentration (r = 0.762, *p* = 0.028). The serum exosomal hsa-miR-145-5p content positively correlated with total cholesterol levels in the entire cohort (r = 0.530, *p* = 0.006).

## 3. Discussion

Here, for the first time, we applied an original approach searching for new biomarkers for the purpose of monitoring T2DM development in obesity. We analyzed the obesity-associated miRNA profile of VAT-derived EVs and then assessed these obesity-specific miRNAs in serum EVs, evaluating their association with obesity and T2DM. hsa-miR-1246, hsa-miR-145-5p, and hsa-miR-551b-3p were shown to be downregulated in VAT EVs from obese individuals. Further elective screening of these microRNAs in serum EVs demonstrated downregulation of hsa-miR-551b-3p in obese individuals without T2DM, as compared to controls. These results are consistent with results from several previous studies showing that the hsa-miR-551b-3p content was reduced in adipose tissue in obesity [24,25]. Additionally, Liu et al. showed a reduction in plasma hsa-miR-551b-3p in obesity-associated metabolic syndrome [26]. We, for the first time, demonstrated a decreased level of hsa-miR-551b-3p, both in VAT and in serum EVs, in obese patients, supporting the assertion that this exosomal miRNA’s secretion could be linked to adipose tissue remodeling in obesity.

Unexpectedly, this miRNA was elevated in serum EVs from T2DM patients and returned to the level of the control group but was still negatively correlated with BMI. Thus, the level of serum exosomal hsa-miR-551b-3p in obese T2DM patients was rather variable. The reason for the observed increase in hsa-miR-551b-3p content in serum EVs from T2DM patients is not clear. It can be assumed that it could be dependent on the disease duration, concomitant complications of T2DM, or antihyperglycemic treatment [27,28]. For example, it was shown that hsa-miR-551b-3p was downregulated in heart tissue during diabetic cardiomyopathy [29,30]. Feng et al. previously demonstrated that hsa-miR-551b-5p is involved in diabetic cardiomyopathy in model animals [31]. Increased miR-551b expression was shown to lead to endothelial dysfunction by upregulating Egr-1 expression [32]. However, we could not exclude the possibility that the rise in the serum exosomal hsa-miR-551b-3p level in T2DM is a consequence of treatment, as the commonly used oral antihyperglycemic agent metformin was shown to alter microRNAs in a hepatocarcinoma cell line [28].

As for hsa-miR-302d-3p, which also was upregulated in T2DM serum EVs in our study, its elevation in circulation in nonobese T2DM patients was earlier shown by Khan et al. [33]. It should be noted that hsa-miR-302d-3p plays an important role in the differentiation of fat cells [34].

Katayama et al. examined the miRNA expression profile of total serum and serum-derived exosome-enriched EVs in men with normal glucose tolerance or T2DM [21]. Unlike in our study, all the participants were slightly overweight men, and patients with a high degree of obesity were not included. After screening a panel of 179 microRNAs in subgroups of four men with normal glucose tolerance and four men with T2DM, upregulations in serum EVs miR-20b-5p and miR-150-5p were detected and then validated in larger cohorts [21]. Still, these miRNAs did not allow for distinguishing impaired glucose tolerance or metabolic syndrome from T2DM, making them unsuitable for T2DM monitoring [21,35]. Interestingly, comparing the total serum microRNA from the same individuals did not reveal any significant differences between men with normal glucose tolerance and those with T2DM, suggesting that exosomal, rather than serum-derived, miRNAs are altered in T2DM [21].

Kim et al., using an NGS approach, showed that the general miRNA profiles of serum exosomes of obese patients with T2DM were similar to those of obese patients without T2DM [20]. But some microRNAs were associated with T2DM: the expression of miR-23a-5p and miR-6087 increased in patients with obesity without T2DM compared with the control group and in patients with obesity with T2DM compared with patients with obesity without T2DM. In contrast, miR-6751-3p was consequentially downregulated [20]. Still, a small number of patients were included, and real-time PCR validation was not performed. No data showed that these microRNAs were different in EVs secreted by adipose tissue [23,36].

It is worth noting that the size distribution and concentration of EVs in serum from men with T2DM or normal glucose tolerance were not different [21]. We earlier analyzed the adipose tissue EV structure by means of cryo-electron microscopy and found no size or morphological differences between such EVs from obese patients with and without T2DM [14]. Observed differences such as the increased total proportion of EVs with internal membrane structures were likely due to obesity-associated disturbance of EV biogenesis [14]. The amount of EVs secreted by obese adipose tissue increases, and their influence on gene expression in target tissues is supposed to be greater, which can result in the development of obesity-related comorbid diseases [23]. Thus, adipose tissue EVs’ composition is more likely to be linked to obesity complications and could be modified by treatment. Hubal et al. studied the association of changes in the miRNA profile of serum adipocyte-derived exosomes one year after bariatric surgery with improving glucose regulation. They showed that miRNAs with target genes within the Insulin Receptor Signaling canonical pathway were associated with a change in the HOMA index [22].

It is important that adipose tissue is a main source of serum exosomal miRNAs, underlining them as good markers for monitoring adipose tissue dysfunction and obesity-associated metabolic diseases. Adipose tissue EVs are primarily of adipocyte origin [23]. Mice with a fat-specific knockout of the miRNA-processing enzyme Dicer (ADicerKO) demonstrated a pronounced decrease in circulating exosomal miRNAs levels: 88% of serum exosomal miRNAs in ADicerKO mice were reduced by >4-fold [5]. Fat transplantation restored the levels of the majority of these miRNAs to at least 50% of normal. At the same time, the reduction in total miRNAs in ADicerKO serum was not so dramatic [5]. Profiling of serum exosomal miRNA from patients with congenital generalized lipodystrophy and patients with HIV-related lipodystrophy, previously shown to have decreased levels of Dicer in adipose tissue, also showed a reduction in microRNA quantity. Therefore, the authors concluded that the majority of human circulating exosomal miRNAs originate from adipose tissue. However, other tissues also contribute to the serum exosomal miRNA profile, which makes it difficult to find relevant biomarkers.

Thus, our study demonstrated a downregulation of exosomal hsa-miR-551b-3p in obesity that was detected both in VAT and in serum EVs. As the serum exosomal hsa-miR-551b-3p level was significantly higher in obese patients with T2DM when compared with obese patients without T2DM, we suggest that its level could be viewed as a potential biomarker of T2DM development in obesity.

However, this study had the limitation of a small sample size.

## 4. Materials and Methods

### 4.1. Study Participants

Patient groups comprised obese subjects with or without T2DM who underwent bariatric surgery and had a body mass index (BMI) of >35. T2DM diagnosis was based on clinical and laboratory characteristics as per the 1999 WHO criteria for diabetes classification and diagnosis [37]. Patients with the following characteristic were included: fasting plasma glucose levels of ≥7.0 mmol/L or 2 h post-challenge glucose levels in an oral glucose tolerance test of ≥11.1 mmol/L. The control group comprised normoglycemic subjects without obesity or T2DM who were selected from a convenience sample of patients undergoing unrelated abdominal procedures. Patient data are presented in Table 1.

### 4.2. Serum Collection

Blood was collected from healthy and diabetic subjects fasted for 12 h and allowed to clot for 30 min. Tubes were centrifuged for 10 min at 1000–1300 g, and serum was collected in RNase free tubes and stored at −80 °C until further processing.

### 4.3. Adipose Tissue Cultivation

Visceral and subcutaneous adipose tissue samples (VAT and SAT) (1–2 g) were excised during surgery from the omentum and the anterior abdominal wall incision site, respectively, immediately placed into Hank’s solution, and transported to the laboratory. VAT and SAT samples were washed with phosphate-buffered saline (PBS), cut into 1–4 mm pieces, transferred to Petri dishes containing DMEM/F12 medium with 10% EV-free serum (Fetal Bovine Serum, exosome-depleted, Thermo Fisher Scientific, Waltham, MA, USA, A2720803) supplemented with 1% gentamicin, and incubated for 12 h. The culture supernatant was prepared via serial centrifugations and filtration. Specifically, it was centrifuged at +4 °C at 300× *g* for 10 min, at 3500 g for 30 min, and at 10,000× *g* for 30 min; afterwards, it was filtered through a 0.22 syringe PES filter to remove lipids, cells, and cellular debris before ultracentrifugation. Culture medium samples were frozen in liquid nitrogen and stored at −80 °C until further processing.

### 4.4. Western Blotting

EVs were isolated from the culture medium or serum using the appropriate Total Exosome Isolation Reagent (Thermo Fisher Scientific, Waltham, MA, USA) according to the manufacturer’s instructions. EVs were lysed 1:1 in ice-cold RIPA buffer containing 50 mMTris-HCl (pH 8.0), 150mM NaCl, 1% Triton X-100, 0.5% sodium deoxycholate, 0.1% SDS, and protease inhibitor cocktail (Roche, Basel, Switzerland). The lysate was centrifuged at 14,000× *g* for 15 min at 4 °C, and the supernatant was carefully aspirated into a new tube. Protein concentrations were determined using the Micro BCA protein assay (Thermo Fisher Scientific, Waltham, MA, USA). A quantity of 5 µg protein per lane was separated using 8% SDS-PAGE gels. Proteins were transferred to PVDF membranes (Millipore, Burlington, MA, USA) and pre-incubated with 5% skim milk in PBS. The blots were incubated with rabbit polyclonal anti-CD63 (1:1000; ab216130, Abcam, Cambridge, UK), anti-FABP4 (1:1000; PA5-30591, Thermo Fisher Scientific, Waltham, MA, USA), anti-omentin-1 (1:2000; AB10627, Merck, Burlington, MA, USA), anti-adiponectin (1:1000, Almabion, Voronezh, Russia), and monoclonal anti-CD81 (1:1000; ab109201, Abcam, Cambridge, UK) primary antibodies diluted in 1% skim milk in PBST (0.05% Tween 20 PBS) to prevent non-specific binding, followed by anti-rabbit HRP-conjugated secondary antibodies (1:3000; ab6721, Abcam, Cambridge, UK). Proteins were visualized using ECL Western Blotting Detection Reagent (Thermo Fisher Scientific, Waltham, MA, USA) and a ChemiDoc Imaging system (BioRad, Hercules, CA, USA).

### 4.5. NanoString

Adipose tissue EVs were isolated from 6 ml of culture medium using the Total Exosome Isolation Reagent (Thermo Fisher Scientific, Waltham, MA, USA). RNA isolation was performed using the Qiazol reagent (Qiagen, Venlo, the Netherlands), followed by purification and concentration using an RNA Clean and Concentrator kit (Zymo Research, Irvine, CA, USA, R1013). The RNA concentration was determined on a Qubit fluorimeter using the RNA HS Assay Kit (Thermo Fisher Scientific, Waltham, MA, USA). MicroRNA profiling was performed via the NanoString method using 10 ng of RNA and the Human v3 miRNA expression assay kit on an nCounter FLEX instrument (NanoString, Seattle, VA, USA).

### 4.6. NanoString Data Normalization and Analysis

Raw data were analyzed using nSolver (v4.0, Nanostring, Seattle, VA, USA). The background level was defined as 2 standard deviations (SD) above the mean negative control counts. Then, the first step of data normalization against ligation controls was performed using nSolver v4.0. Next, the two endogenous miRNA controls (hsa-miR-378g and hsa-let-7c-5p) were identified from the dataset using the NormFinder algorithm, which is used to select the best stable single reference gene or combination thereof. NormFinder incorporates the inter- and intra-group variances and calculates the stability value per target as a measure of expression stability, and then it identifies the best combination of targets to provide the highest stability [38]. Thus, in the second step, data were normalized against the geometric mean of the combination of hsa-miR-378g and hsa-let-7c-5p selected by NormFinder from the whole set of miRNAs detected in adipose tissue EVs. Further, we used the limma package (version 3.50.0) for differential expression [39,40]. We converted the normalized read counts using the voom function of limma, fitted the linear model with the lmFit function, and calculated *p*-values using the eBayes function [41]. We considered miRNAs differentially expressed with an FDR adjusted *p*-value of <0.05 and absolute log2 fold change between conditions of >1. We carried out a number of comparisons: between VAT EVs of obese patients and controls; between SAT and VAT EVs of obese patients.

### 4.7. GO and Pathway Enrichment Analysis

The analysis was carried out using the miRTargetLink 2.0 online resource [42]. This resource allows the identification of target genes and visualization of interaction patterns between selected miRNAs and genes. To identify potential target genes for miRNAs differentially expressed in EVs, we considered both experimentally verified interactions and those predicted using bioinformatic programs, limiting the number of target genes to interactions with at least 2 miRNAs. More than 2500 genes have been identified. The online tool Database for Annotation, Visualization and Integrated Discovery (DAVID) (version 6.8) was used to detect GO (Gene Ontology) categories and KEGG (Kyoto Encyclopedia of Genes and Genomes) pathways [43].

### 4.8. Serum Exosomal miRNA Analysis

EVs were isolated from 0.5 mL of serum using the Total Exosome Isolation Reagent (Thermo Fisher Scientific, Waltham, MA, USA). RNA isolation was performed using the Qiazol reagent (Qiagen), followed by purification and concentration using an RNA Clean and Concentrator kit (Zymo Research, Irvine, CA, USA, R1013). miRNAs were detected using stem-loop real-time PCR according to the method offered by Lan et al. [44]. Reverse transcription (RT) was performed using miRNA-specific RT primers and an MMLV kit (Eurogen, Moscow, Russia) using 2 µL of exosomal RNA sample; thus, miRNAs of interest were specifically elongated during reverse transcription. Quantitative real-time PCR analysis was conducted using the CFX96 Real-Time PCR Detection System (Bio-Rad, Hercules, CA, USA) with HS-SYBR master-mix (Eurogen, Moscow, Russia). Data were normalized to 2 reference microRNAs (has-miR-378g ahashsa-let-7c-5p), the same as in the Nanostring experiment, using the 2–ΔCt method and expressed as relative units. The primer sequences used for RT and real-time PCR are listed in Appendix A.

### 4.9. Statistical Analysis

Clinical and experimental data are presented as the mean ± standard deviation (SD) or the median (min–max) depending on the distribution. The distribution of continuous variables was tested for normality using the Shapiro–Wilk test. For non-normally distributed variables, comparisons between two groups were performed using the Mann–Whitney U test, while those among three groups were performed using the Kruskal–Wallis and Dunn’s post hoc tests. The sensitivity and specificity of potential biomarkers were assessed via ROC analysis. Correlations between variables were tested by using a two-tailed Spearman’s test. The level of significance was set at *p* < 0.05. A description of the NanoString data analysis is presented in the corresponding subsection.

## Figures and Tables

**Figure 2 ncrna-09-00067-f002:**
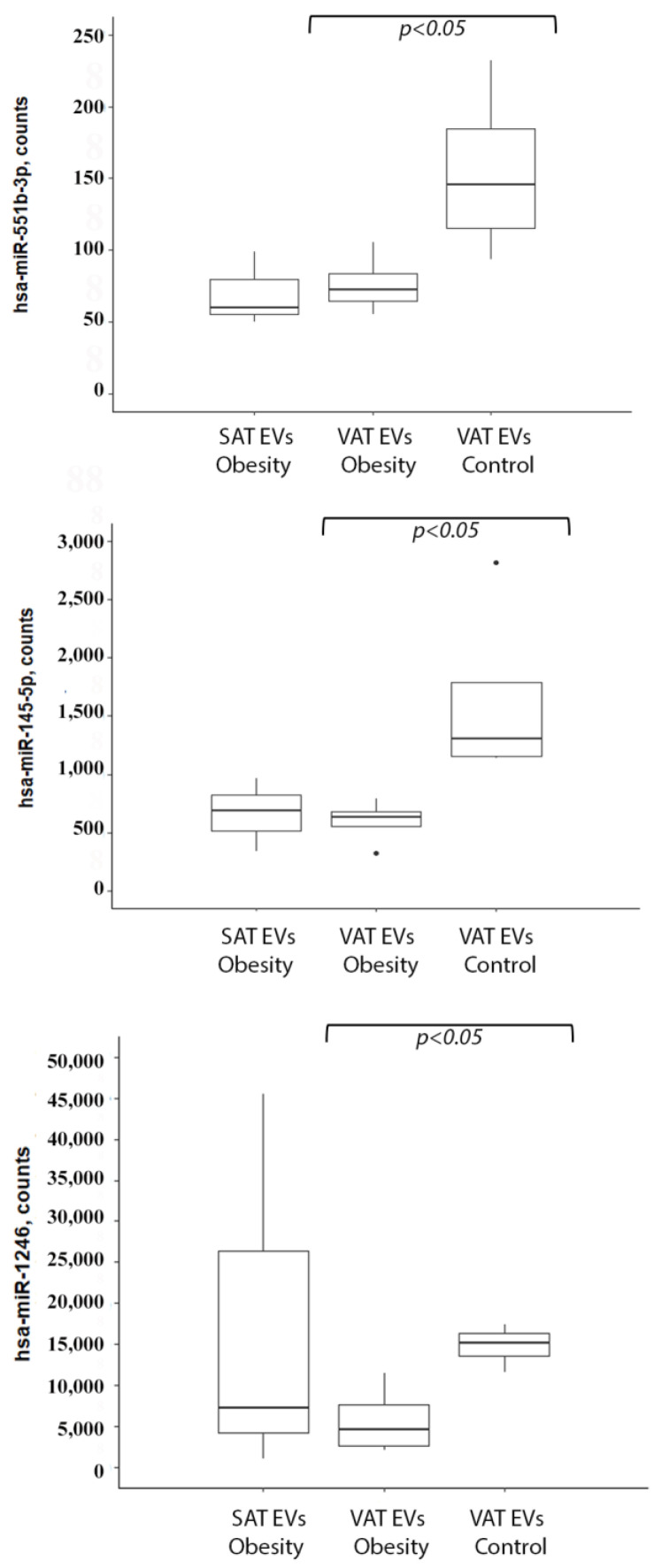
miRNAs downregulated in VAT EVs in obesity. Abbreviations: EVs—extracellular vesicles, SAT—subcutaneous adipose tissue, VAT—visceral adipose tissue.

**Figure 3 ncrna-09-00067-f003:**
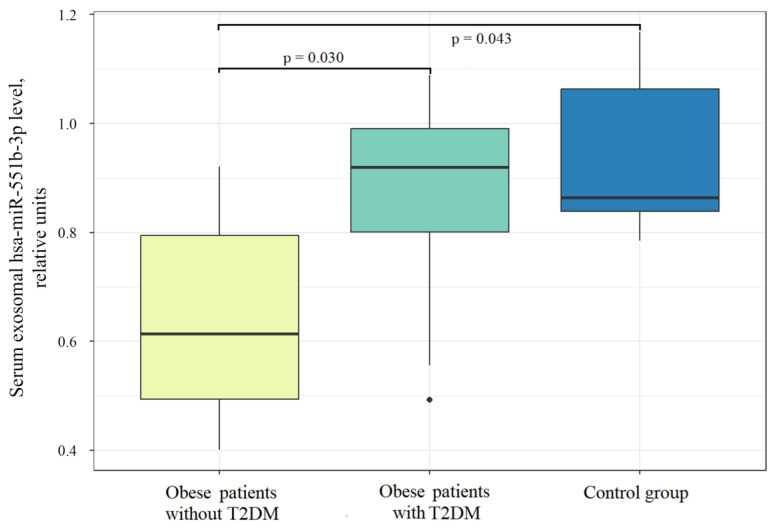
hsa-miR-551b-3p abundance in serum exosomal RNA in the studied groups.

**Figure 4 ncrna-09-00067-f004:**
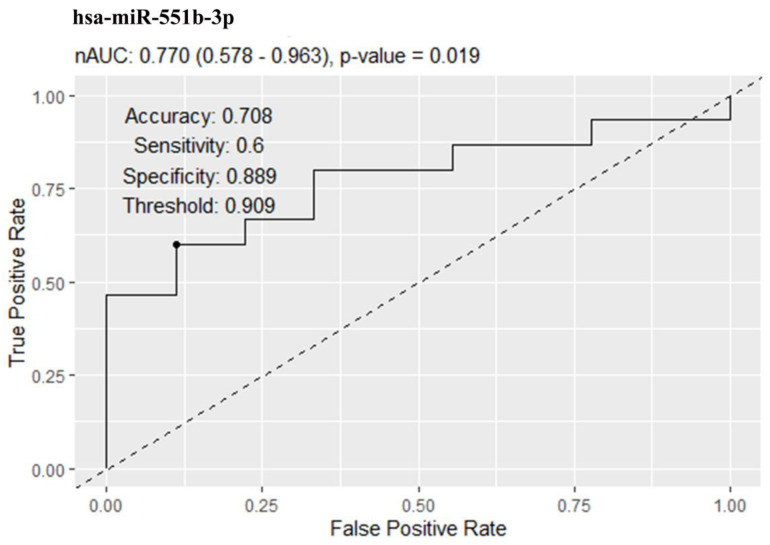
Results of sensitivity and specificity analysis of the serum exosomal hsa-miR-551b-3p level as a marker of T2DM development in obesity.

**Figure 1 ncrna-09-00067-f001:**
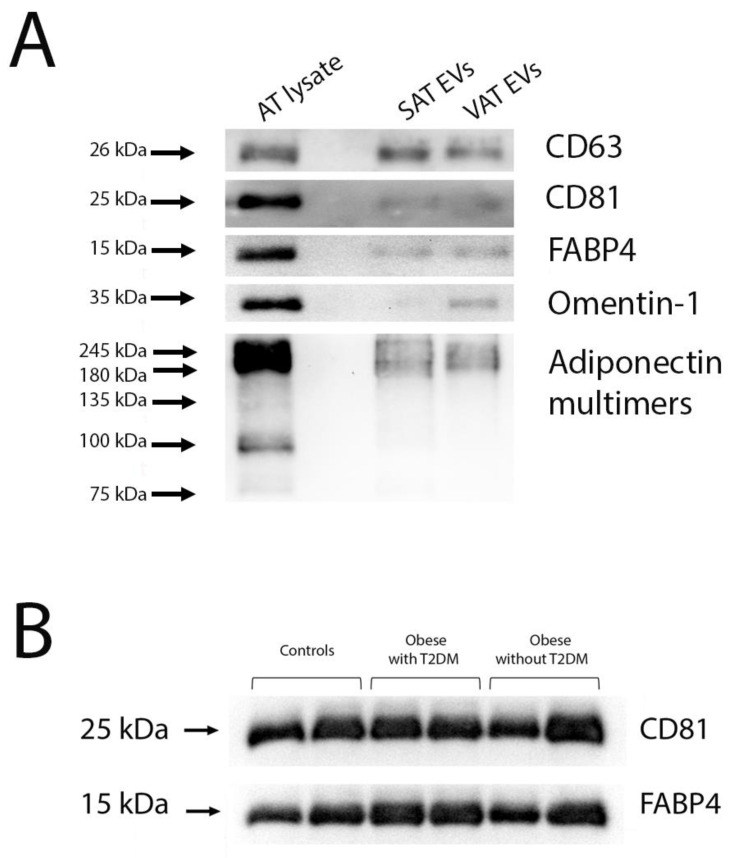
Representative Western blots of adipose tissue EVs (**A**) and serum EVs (**B**). Abbreviations: AT—adipose tissue, EVs—extracellular vesicles, SAT—subcutaneous adipose tissue, VAT—visceral adipose tissue.

**Table 1 ncrna-09-00067-t001:** Demographic, anthropometric, and biochemical data of patients included in the study.

Groups/ Parameter	Obesity without T2DM *n* = 9	Obesity with T2DM *n* = 15	Control Group *n* = 7	*p*
Age, years	41.0 ± 14.7	47.9 ± 10.9	45.3 ± 4.9	^1^0.215 ^2^0.781 ^3^0.527
Sex (male/female)	2/7	6/9	2/5	
Weight (kg)	118.6 ± 18.1	140.8 ± 21.4	66.0 ± 8.5	**^1^0.046** ^2^ **0.021** **^3^0.008**
Body mass index, kg/m^2^	41.9 ± 5.6	48.9 ± 6.1	24.9 ± 2.8	**^1^0.016** ^2^ **0.017** **^3^0.006**
Waist circumference, cm	107.0 (99.0–143.0)	145.0 (123.0–160.0)	nd	**^1^0.002**
Hip, cm	126.5 (100.0–141.0)	135 (115.0–145.0)	nd	^1^0.159
Waist-to-hip ratio	0.9 ± 0.2	1.1 ± 0.1	nd	**^1^0.027**
Glucose, nmol/L	5.4 (4.3–8.1)	7.3 (5.5–12.7)	4.9 (4.3–5.0)	**^1^0.007** ^2^0.389 **^3^0.007**
Insulin, µIU/mL	26.86 (9.98–41.4)	28.1 (16.3–79.4)	nd	^1^0.907
HOMA-IR index	6.1 ± 2.7	10.7 ± 5.4	nd	^1^0.165
C-peptide (ng/mL)	3.5 (1.8–4.6)	3.8 (1.9–11.9)	nd	^1^0.587
HbA1c, %	5.6 ± 0.2	7.45 ± 1.7	nd	**^1^0.010**
Total cholesterol, mmol/L	4.9 ± 1.4	5.1 ± 0.8	nd	^1^1.000
HDL cholesterol, mmol/L	1.6 ± 0.3	1.2 ± 0.2	nd	^1^0.182
LDL cholesterol, mmol/L	2.9 ± 1.4	2.7 ± 0.9	nd	^1^0.570
Triglyceride, mmol/L	1.7 (1.2–2.9)	2.2 (1.3–4.4)	nd	^1^0.101

Notes: nd—not determined; ^1^ obese T2DM vs. non-T2DM, ^2^ non-T2DM obese vs. control, ^3^ T2DM vs. control. Abbreviations: T2DM—type 2 diabetes mellitus, HOMA-IR—homeostasis model assessment of insulin resistance, HDL—high-density lipoproteins, LDL—low-density lipoproteins.

## Data Availability

Not applicable.

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
