# Peer review of "Downregulation of Exosomal hsa-miR-551b-3p in Obesity and Its Link to Type 2 Diabetes Mellitus"

_ncrna, 2023, doi:10.3390/ncrna9060067_

Round 1

Reviewer 1 Report

Comments and Suggestions for Authors

The study by Kseniia V. Dracheva and her/his colleagues studied obesity-associated miRNA profile of VAT-derived EVs and then assessed these obesity-specific miRNAs in serum EVs evaluating their association with obesity and T2DM. The topic of the study is of high importance since diabetes and obesity are considered as major public health problem with increasing prevalence worldwide.

There are some concerns:

1.     What does it mean (BMI>35) and (BMI<30)? (line 15)., and AT (line 26)?

2.     In table 1, age was presented. What is the unit of age? Months? Year? Or any? Please mention it. Please check the spelling of the unit of BMI. The mean value of BMI for the control group is almost on the threshold (normal weight), which is 24.9. So, can we say it is the control group?  I do not think so. I think there is a problem with the selection of a control group. I wonder if you could justify it. There was a significant statistical association between weight (118.6±18.1) and (140.8±21.4), however, no significant association was observed between (118.6±18.1) and (66.0±8.5). how and why? Could you please justify it? The same was also observed in BMI and other parameters as well. Please check them.

3.     The study participants are very few. Could you please increase the number of study participants if it is possible?

4.     The study design is not easy for readers to understand. I strongly suggest if you add a figure, which summarizes the study design.

Author Response

We thank the reviewer for the attention paid to our work, for showing us some typos that we were able to correct, and for irreplaceable advices on how best to present the work. Here we consistently respond to all comments point by point.

  1. What does it mean (BMI>35) and (BMI<30)? (line 15)., and AT (line 26)?

These are undeciphered abbreviations of body mass index and adipose tissue, we now corrected this misunderstanding.

  1. In table 1, age was presented. What is the unit of age? Months? Year? Or any? Please mention it. Please check the spelling of the unit of BMI. The mean value of BMI for the control group is almost on the threshold (normal weight), which is 24.9. So, can we say it is the control group?  I do not think so. I think there is a problem with the selection of a control group. I wonder if you could justify it. There was a significant statistical association between weight (118.6±18.1) and (140.8±21.4), however, no significant association was observed between (118.6±18.1) and (66.0±8.5). how and why? Could you please justify it? The same was also observed in BMI and other parameters as well. Please check them.

We clarify what is the control group and criteria for diagnosing obesity in the Methods section: “Control group was formed by normoglycemic subjects without obesity and T2DM who was selected from a convenience sample of patients undergoing unrelated abdominal procedures. Patient data is represented in table 1.” So, control group is presented by individuals without obesity diagnosis. Maybe they are slightly overweight, but this is a real problem to get enough material for adipose tissue cultivation from thin patients. This shouldn't be a problem because additionally there is a rather wide “BMI window” 30-35. Also, for this point - units were clarified and typos in p values were corrected.

  1. The study participants are very few. Could you please increase the number of study participants if it is possible?

Unfortunately, we are not able to enlarge the number of participants now because we are not able to get the same reagent kits and alll samples must be proceeded in the same manner. We will indicate small sample sizes as the limitation of the study. Also, we can note that there are difficulties with collection enough material for cultivation of adipose tissue.

  1. The study design is not easy for readers to understand. I strongly suggest if you add a figure, which summarizes the study design.

We agree and want to add a graphical abstract.

Reviewer 2 Report

Comments and Suggestions for Authors

In this manuscript the authors examined exosomal miRNAs in obesity and type 2 diabetes patients.  They found that in particular miR-551b-3p might be an EV biomarker for T2DM in obesity.  Two main issues about the manuscript are noted below.

1. The sample numbers are low (Table 1).

2. How many miRNAs were initially screened for the differences?  Are they all listed in Table S1?  If so, what was the basis for their selection? What are “hsa-378g” and “has-7c-5p”, and why were they used as controls?  Figure 2 and S1 indicated that a significant number of these miRNAs were differentially present in various EVs.  A small sample size, a limited number of tests (miRNAs), and no appropriate internal controls might give spurious results.

Comments on the Quality of English Language

The manuscript needs editing.  Just a few examples here.  In the title, “it’s” should be “its”, and "link with" better be "link to" or "association with".  In Discussion, “It worst to noting” should be “It is worth noting”.  Define or correct “hsa-378g” and “has-7c-5p”.

Author Response

We thank the reviewer for the attention paid to our work, for showing us some typos that we were able to correct, and for irreplaceable advices on how best to present the work. Here we consistently respond to all comments point by point.

  1. The sample numbers are low (Table 1).

Unfortunately, we are not able to enlarge the number of participants now because we are not able to get the same reagent kits and all samples must be proceeded in the same manner. We will indicate small sample sizes as the limitation of the study. Also, we can note that there are difficulties with collection enough material for cultivation of adipose tissue.

  1. How many miRNAs were initially screened for the differences?  Are they all listed in Table S1?  If so, what was the basis for their selection? What are “hsa-378g” and “has-7c-5p”, and why were they used as controls?  Figure 2 and S1 indicated that a significant number of these miRNAs were differentially present in various EVs.  A small sample size, a limited number of tests (miRNAs), and no appropriate internal controls might give spurious results.

We tried to make detailed description of methods in the corresponding section (lines 290-313). At first stage NanoString experiment was established to find differently presented miRNAs in EVs secreted by adipose tissue. We used miRNA panel which includes 800 miRNAs (Now we add information on the number of miRNA in NanoString experiment to Results section too “Human v3 miRNA expression assay kit allows simultaneous detection of 800 miRNAs” line 108-109). There are certain features when normalizing and analyzing NanoString miRNA data. RNA samples for miRNA detection are exposed to ligation with specific adapters as miRNA are too small. And only then hybridized with specific probes with molecular barcodes. So, panel includes ligation controls – that’s why at first data were normalized to these controls. After this step stable combination of miRNAs that can be used as internal controls was determined by NormFinder algorithm. These were hsa-378g and hsa-7c-5p. Then data was normalized and differences were estimated by limma voom approach usually used for differential gene expression analysis and suitable for NanoString counts. As we were aimed to find markers that mirror adipose tissue dysfunction and obesity associated differences in miRNA content of secreted EVs it was logic to use the same internal control miRNAs in the serum. So, we see quantity of miRNAs of interest in relation to these controls. That’s why primers for hsa-378g and hsa-7c-5p are presented in the corresponding table.

Reviewer 3 Report

Comments and Suggestions for Authors

This study by Dracheva et al provides an examination of exosomal microRNAs (miRNAs) derived from adipose tissue and serum, shedding light on their potential roles in obesity and Type 2 Diabetes Mellitus (T2DM). The initial study involves characterizing the study cohorts using demographic and anthropometric data. The authors then delve into the miRNA composition of adipose tissue-derived extracellular vesicles (EVs) in ex vivo cultures. This process is crucial for understanding the molecular dynamics within adipose tissue. Additionally, the study extends its scope to investigate the content of obesity-associated miRNAs in serum EVs, stratifying patients based on T2DM diagnosis. One notable strength of this study is the utilization of NanoString technology for miRNA profiling, which offers a high-throughput and sensitive platform for this purpose. The results reveal specific miRNAs that are downregulated in visceral adipose tissue (VAT) EVs of obese individuals, providing valuable insights into the molecular mechanisms associated with obesity. The subsequent functional and pathway enrichment analyses, conducted using the DAVID database, reveal a connection between the identified miRNAs and pathways related to lipid metabolism, adipose tissue cell differentiation, and insulin signaling. This suggests a potential link between these miRNAs and the development of metabolic complications in obesity, including T2DM. The study's extension into serum EVs in a larger patient cohort provides further findings. The reduction of hsa-miR-551b-3p in serum exosomal content in obese patients without T2DM compared to the control group is notable. The correlations between exosomal miRNA levels and clinical parameters are important. hsa-miR-551b-3p is positively correlated to glycated hemoglobin (HbA1c) but negatively with BMI in the T2DM subgroup. In conclusion, this study investigates the miRNA composition of adipose tissue and serum EVs, illuminating potential links to obesity and T2DM. The use of advanced technology, data analysis, and insightful correlations with clinical parameters contribute to the overall strength of the research. 

I have a few minor concerns that can be easily addressed:

1. Table 1 can be made more visually appealing and the notes section can be improved too.

2. In many places within the text, '.' has been errorenously written as ',' (line 111, 147, 148, 263)

3. In line 113, there is a character that is supposed to be 'and' that can be replaced, and 'were' should be omitted.

Comments on the Quality of English Language

English quality and grammar can be improved.

Author Response

We thank the reviewer for the attention paid to our work, for showing us some typos that we were able to correct, and for irreplaceable advices on how best to present the work. Here we consistently respond to all comments point by point.

  1. Table 1 can be made more visually appealing and the notes section can be improved too.

We agree but as there was a demand to use journal template, we suppose the editors will then format the table so that the rows match. Additionally, according to advice of one of the reviewers we try to add graphical abstract, which summarizes the study design and complement the table.

  1. In many places within the text, '.' has been errorenously written as ',' (line 111, 147, 148, 263)

corrected

3.In line 113, there is a character that is supposed to be 'and' that can be replaced, and 'were' should be omitted.

done

Round 2

Reviewer 2 Report

Comments and Suggestions for Authors

What are hsa-378g and hsa-7c-5p?  Because these are not standard miRNA names.  Even if they are included in the company's kit, they should still be explained in the manuscript.  This will spare readers from BLASTing the primers.

Comments on the Quality of English Language

OK.

Author Response

We thank the reviewer for carefully looking at our manuscript. It helped us to correct mistakes and make more clear description of experiment. Here is the answer for the last comment:

At first we apologize by mistake we missed in miRNA names “miR” and “let”. So, these miRNAs are “hsa-miR-378g and hsa-let-7c-5p”. They were determined by NormFinder algorithm as stable controls (it is important – it’s combination) from the whole set of miRNAs which were detected by NanoString in adipose tissue extracellular vesicles. Also, we agree that it was likely not obvious that these miRNA names appeared in table S1. Now we tried to explain the procedure more carefully in lines 303-310. NormFinder is a common approach to select the best stable single or combination of reference genes from a panel of candidates. So, we use it for our miRNA dataset as there is no consensus regarding the normalization of miRNAs of body fluids and especially for exosomes. Also, we added names “hsa-miR-378g and hsa-let-7c-5p” in PCR subsection in line 337 to emphasize that reference miRNAs are the same.